# TRAINING GENERATIVE LATENT MODELS BY VARIATIONAL $f$-DIVERGENCE MINIMIZATION

## ABSTRACT

Probabilistic models are often trained by maximum likelihood, which corresponds to minimizing a specific form of $f$-divergence between the model and data distribution. We derive an upper bound that holds for all $f$-divergences, showing the intuitive result that the divergence between two joint distributions is at least as great as the divergence between their corresponding marginals. Additionally, the $f$-divergence is not formally defined when two distributions have different supports. We thus propose a noisy version of $f$-divergence which is well defined in such situations. We demonstrate how the bound and the new version of $f$-divergence can be readily used to train complex probabilistic generative models of data and that the fitted model can depend significantly on the particular divergence used.

## 1 INTRODUCTION

Probabilistic modelling generally deals with the task of trying to fit a model $p_\theta(x)$ parameterized by $\theta$ to a given distribution $p(x)$. To fit the model we often wish to minimize some measure of difference between $p_\theta(x)$ and $p(x)$. A popular choice is the class of $f$-divergences[1] (see for example Sason & Verdú (2015)) which, for two distributions $p(x)$ and $q(x)$, is defined by

$$\mathrm{D}_f(p(x)||q(x)) = \int q(x) f\left(\frac{p(x)}{q(x)}\right) dx \qquad (1)$$

where $f(x)$ is a convex function with $f(1) = 0$.

Many of the standard divergences correspond to simple choices of the function $f$, see table 1. For example, for $f(u) = u \log u$ we have the "forward" KL divergence $\mathrm{KL}(p(x)||q(x))$; setting $f(u) = -\log u$ gives the "reverse" KL divergence $\mathrm{KL}(q(x)||p(x))$. The divergence $\mathrm{D}_f(p(x)||q(x))$ is zero if and only if $p(x) = q(x)$. However, for a constrained model $p_\theta(x)$ fitted to a distribution $p(x)$ by minimizing $\mathrm{D}_f(p_\theta(x)||p(x))$, the resulting optimal $\theta$ can be heavily dependent on the choice of the divergence function $f$ (Minka, 2005).

Whilst there is significant recent interest in using $f$-divergences to train complex probabilistic models (Nowozin et al., 2016; Goodfellow et al., 2014), the $f$-divergence is generally computationally intractable for such complex models. The main contribution of our paper is the introduction of an upper bound on the $f$-divergence. We show how this bound can be readily applied to training complex latent variable generative models based on only a modest departure from the standard Variational Autoencoder (Kingma & Welling, 2013).

## 2 BACKGROUND

### 2.1 MAXIMUM LIKELIHOOD

For data $x_1, \ldots, x_N$ drawn independently and identically from some unknown distribution, fitting an approximating distribution $p_\theta(x)$ by minimizing the forward KL between $p_\theta(x)$ and the empirical distribution $\hat{p}(x)$ of the data we have

---

[1]The definition extends naturally to distributions on discrete $x$.

| Name | $D_f(p(x)\|q(x))$ | $f(u)$ |
|---|---|---|
| "forward" Kullback-Leibler | $\int p(x)\log\frac{p(x)}{q(x)}dx$ | $u\log u$ |
| "reverse" Kullback-Leibler | $\int q(x)\log\frac{q(x)}{p(x)}dx$ | $-\log u$ |
| Jensen-Shannon | $\int\left\{\frac{1}{2}p(x)\log\frac{2p(x)}{p(x)+q(x)}+q(x)\log\frac{2q(x)}{p(x)+q(x)}\right\}dx$ | $-(u+1)\log\frac{1+u}{2}+u\log u$ |
| GAN | $\int\left\{p(x)\log\frac{2p(x)}{p(x)+q(x)}+q(x)\log\frac{2q(x)}{p(x)+q(x)}\right\}dx$ | $-(u+1)\log(1+u)+u\log u$ |

Table 1: Some standard $f$-divergences. $p(x)$ is given and $q(x)$ is the model. From Nowozin et al. (2016).

$$\hat{p}(x)\equiv\frac{1}{N}\sum_{n=1}^{N}\delta\left(x-x_n\right)\quad\Rightarrow\quad \text{KL}(\hat{p}(x)\|p_\theta(x))=-\sum_{n=1}^{N}\log p_\theta(x_n)+const. \qquad (2)$$

Minimizing $\text{KL}(\hat{p}(x)\|p_\theta(x))$ w.r.t. $\theta$ is therefore equivalent to maximizing the likelihood of the data, $p_\theta(x)$. Given the asymptotic guarantees of the efficiency of maximum likelihood (Wolfowitz, 1965), the forward KL is the standard divergence used in statistics and machine learning.

## 2.2 FORWARD VERSUS REVERSE KL

It is interesting to compare models trained by the forward and reverse KL divergences. For example, when $p_\theta$ is Gaussian with parameters $\theta=\left(\mu,\sigma^2\right)$, then minimizing the forward KL gives

$$\arg\min_{\mu,\sigma^2}\text{KL}(p(x)\|p_\theta(x))\quad\Rightarrow\quad\mu=\int p(x)xdx,\quad\sigma^2=\int p(x)(x-\mu)^2dx \qquad (3)$$

so that the optimal setting is for $\mu$ to be the mean of $p(x)$ and $\sigma^2$ the variance. For an "under-powered" model (a model which is not rich enough to have a small divergence) $p_\theta(x)$ and multi-modal $p(x)$ this could result in $p_\theta(x)$ placing significant mass on low probability regions in $p(x)$. This is the so-called "mean matching" behavior of $\text{KL}(p(x)\|p_\theta(x))$ that has been suggested as a possible explanation for the poor fidelity of images generated by models $p_\theta(x)$ trained by forward KL minimization (Goodfellow, 2016). Conversely, when using the reverse KL objective $\text{KL}(p_\theta(x)\|p(x))$, for a Gaussian $p_\theta(x)$ and multi-modal $p(x)$ with well separated modes, optimally $\mu$ and $\sigma^2$ fit one of the local modes. This behavior is illustrated in figure 1 and is the so-called "mode matching" behavior of $\text{KL}(p_\theta(x)\|p(x))$. For this reason, the reverse KL objective has been suggested to be useful if high quality data samples are preferable to coverage of the dataset (Goodfellow, 2016).

This highlights the potentially significant difference in the resulting model that is fitted to the data, depending on the choice of divergence (Minka, 2005). In this sense, it is of interest to explore fitting generative models $p_\theta(x)$ to a data distribution $p(x)$ using $f$-divergences other than the forward KL divergence (maximum likelihood).

## 2.3 LATENT GENERATIVE MODELS

For the model $p_\theta(x)$ to have the power to generate complex datasets, we introduce a latent variable $z$, i.e. $p_\theta(x)=\int p_\theta(x|z)p(z)dz$. Following standard practice we do not place any parameters on the prior for the latent $z$, though this would be straightforward.

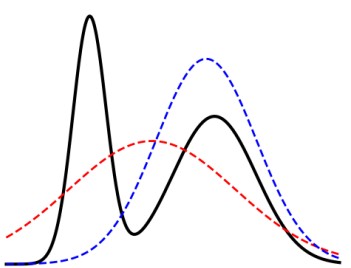

Figure 1: Fitting a Gaussian to a mixture of Gaussians by minimizing the forward KL (red) and the reverse KL (blue).

A standard approach to fitting a latent generative model to data $x_1,\ldots,x_N$ is maximum likelihood.

$$\sum_{n=1}^{N}\log p_\theta(x_n)=\sum_{n=1}^{N}\log\int p_\theta(x_n|z_n)p(z_n)dz_n \qquad (4)$$

In all but simple cases the integral above over the latent $z$ is intractable and a lower bound is used instead

$$\sum_{n=1}^{N} \log p_\theta(x_n) \geq \sum_{n=1}^{N} \int q_\phi(z_n|x_n) \left[\log p_\theta(x_n|z_n)p(z_n) - \log q_\phi(z_n|x_n)\right] dz_n \equiv L(\theta, \phi) \quad (5)$$

where the so-called variational distribution $q_\phi(z|x)$ is chosen such that the bound (and its gradient) is either computationally tractable or can be readily estimated by sampling (Kingma & Welling, 2013). The parameters $\phi$ of the variational distribution $q_\phi(z|x)$ and parameters $\theta$ of the model $p_\theta(x)$ are jointly optimized to increase the log-likelihood lower bound $L(\theta, \phi)$. This lower bound on the likelihood corresponds to an upper bound on the forward divergence $\text{KL}(\hat{p}(x)||p_\theta(x))$, where $\hat{p}(x)$ is the empirical data distribution.

Our interest is to train latent generative models using a different divergence from the forward KL.

Whilst the above upper bound (5) on the forward divergence $\text{KL}(\hat{p}(x)||p_\theta(x))$ is well known, an upper bound on other $f$-divergences seems to be unfamiliar (Sason & Verdú, 2015) and we are unaware of any upper bound on general $f$-divergences that has been used within the machine learning community.

Recently a *lower bound* on the $f$-divergence was introduced in Nowozin et al. (2016) by the use of the Fenchel conjugate. The resulting training algorithm is a form of minimax in which the parameters $\phi$ that tighten the bound are adjusted so as to push up the bound towards the true divergence, whilst the model parameters $\theta$ are adjusted to lower the bound. In Nowozin et al. (2016) the authors were then able to relate the Generative Adversarial Network (GAN) (Goodfellow, 2016) training algorithm to the Fenchel conjugate lower bound on a corresponding $f$-divergence, see table 1. In contrast, if the interest is purely on minimizing an $f$-divergence, it is arguably preferable to have an *upper bound* on the divergence since then standard optimization methods can be applied, resulting in a stable optimization procedure, see figure 2.

## 2.4 $f$-DIVERGENCE BETWEEN DISTRIBUTIONS WITH DISJOINT SUPPORTS

The $f$-divergence between two distributions, $\text{D}_f(p(x)||p_\theta(x))$, is well defined when $p(x)$ and $p_\theta(x)$ have the same support. Many datasets that we are interested in, e.g. images, are generally believed to lie on low dimension manifolds embedded in high-dimensional space (Narayanan & Mitter (2010)). Moreover, if our model is a mixture of delta functions, then the support of $p_\theta(x)$ is contained in a countable union of manifolds of dimension at most dim $\mathcal{Z}$. In this case, the supports of the data distribution and the model distribution are disjoint, so the $f$-divergence is not formally defined, we refer readers to Arjovsky & Bottou (2017) for details. To solve the problem, Sønderby et al. (2017) proposed the instance noise trick and applied it to stabilize the training of the discriminator in GANs. Barber et al. (2018) introduced and discussed the relevant properties of a divergence which is well defined on distributions with different supports. We extend this idea to construct a surrogate of $f$-divergence in section 3.1 and show how to train the new divergence using the auxiliary upper bound.

## 3 THE $f$-DIVERGENCE UPPER BOUND

A central contribution of our paper is the following upper bound on $\text{D}_f(p(x)||q(x))$ between any two distributions $p(x)$ and $q(x)$

$$\text{D}_f(p(x,z)||q(x,z)) = \int q(x,z)f\left(\frac{p(x,z)}{q(x,z)}\right) dx dz \geq \text{D}_f(p(x)||q(x)) \quad (6)$$

where $p(x,z)$ is a distribution with marginal $\int p(x,z)dz = p(x)$ and similarly $\int q(x,z)dz = q(x)$. The bound corresponds to a generalization of the auxiliary variational method (Agakov & Barber,

2004) and follows from a straightforward application of Jensen's inequality:

$$
\begin{aligned}
\mathrm{D}_f(p(x,z)||q(x,z)) &= \int q(x) \int q(z|x) f\left(\frac{p(x,z)}{q(x,z)}\right) dz dx \\
&\geq \int q(x) f\left(\int q(z|x)\frac{p(x,z)}{q(z|x)q(x)}dz\right) dx \\
&= \int q(x) f\left(\frac{p(x)}{q(x)}\right) dx = \mathrm{D}_f(p(x)||q(x))
\end{aligned}
$$

The result states that the divergence between two joint distributions is no less than the divergence between their marginals. Additional properties of the auxiliary $f$-divergence are given in section A of the supplementary material. We show that the bound is tight and reduces to $\mathrm{D}_f(p(x)||q(x))$ when performing a full unconstrained minimization of the bound with respect to $p(z|x)$ (keeping $q(x,z)$ fixed).

## 3.1 Spread $f$-divergence

Recently, Barber et al. (2018) proposed the spread divergence. We give a brief introduction and show how to apply the spread divergence to a $f$-divergence in order to alleviate the problem of disjoint supports between the data and model distributions, as discussed in section 2.4.

For $q(x)$ and $p(x)$ which have disjoint supports, we define new distributions $q(y)$ and $p(y)$ that have the same support. We let

$$
p(y) = \int_x p(y|x)p(x) \qquad q(y) = \int_x p(y|x)q(x) \tag{7}
$$

where $p(y|x)$ is a "noise" process designed such that $p(y)$ and $q(y)$ have the same support. For example, if we use a Gaussian $p(y|x) = \mathcal{N}\left(y|x, \sigma^2\right)$, then $p(y)$ and $q(y)$ both have support $\mathbb{R}$.

We thus define the *spread $f$-divergence*

$$
D'_f(q(x)||p(x)) = D_f(q(y)||p(y)) \tag{8}
$$

This satisfies the requirements of a divergence, that is $D'_f(q(x)||p(x)) \geq 0$ and $D'_f(q(x)||p(x)) = 0$ if and only if $q(x) = p(x)$.

The auxiliary upper bound can be easily applied to the spread $f$-divergence

$$
D'_f(p(x)||q(x)) = D_f(p(y)||q(y)) \leq D_f(p(y,z)||q(y,z)) \tag{9}
$$

## 3.2 Training latent generative models

The bound (9) can be directly applied to form an upper bound on $f$-divergences for training latent generative models, including implicit models with deterministic output $p_\theta(x,z) = p_\theta(x|z)p(z) = \delta(x - \mu_\theta(z))p(z)$. The key idea is that, in situations in which the divergence $\mathrm{D}_f(p(x)||p_\theta(x))$ is intractable, the spread divergence of the joint $\mathrm{D}_f(p(y,z)||p_\theta(y,z))$ may be tractable. We introduce a variational distribution $q_\phi(z|y)$ to express the joint $p(y,z) = q_\phi(z|y)p(y)$. Our generative model factorizes $p_\theta(y,z) = p_\theta(y|z)p(z)$ as before. The bound is then

$$
\mathrm{D}_f(p(y)||p_\theta(y)) \leq \mathrm{D}_f(q_\phi(z|y)p(y)||p_\theta(y|z)p(z)) \equiv U(\theta,\phi) \tag{10}
$$

Once the model has been trained we can recover the model on $x$ by inverting the noise process. We will use a Gaussian noise process, $p(y|x) = \mathcal{N}\left(y|x, \sigma^2\right)$, and a Gaussian output for the generative model, $p_\theta(y|z) = \mathcal{N}\left(y|\mu_\theta(z), \sigma^2\right)$, with both set to use the same fixed variance (which allows us to invert the noise process straightforwardly).

So in practice, we learn a generative model on the $y$-space, $p_\theta(y)$, by minimizing the $f$-divergence to the noise corrupted data distribution $p(y)$. After training we then recover the generative model on $x$-space, $p_\theta(x)$, by taking the mean of our generation network $p_\theta(y|z)$ model as the output. Note that this model $p_\theta(x)$ is still stochastic (i.e. not deterministic) because of the stochasticity of the latent variable $p_\theta(x) = \int \delta(x - \mu_\theta(z))p(z)dz$. By using the spread divergence we have ensured both that

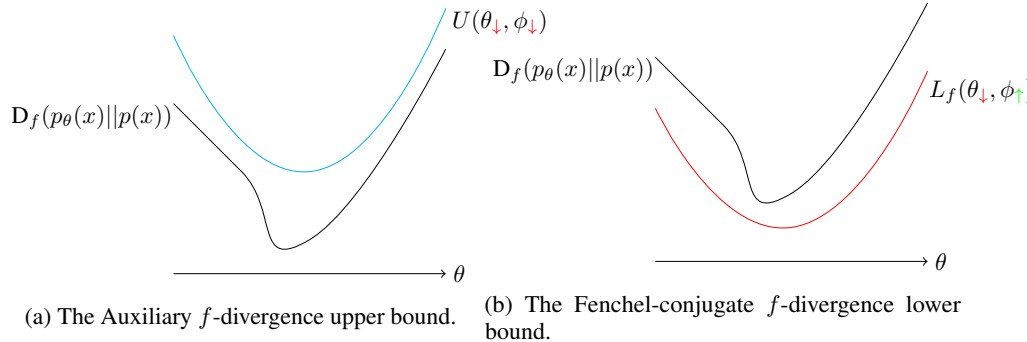

(a) The Auxiliary $f$-divergence upper bound.

(b) The Fenchel-conjugate $f$-divergence lower bound.

Figure 2: Upper and lower bounds on the divergence $\mathrm{D}_f(p_\theta(x)||p(x))$. In our upper bound, both the model parameters $\theta$ and bound parameters $\phi$ are adjusted to push down the upper bound, thereby driving down the divergence. In the Fenchel-conjugate approach Nowozin et al. (2016), the lower bound is made tighter adjusting the bound parameters $\phi$ to push up the bound towards the true divergence, whilst then minimizing this with respect the model parameters $\theta$.

our divergence is properly defined and that we have a way to invert the noise process in a theoretically grounded manner.

Similar to standard treatments in variational inference (see for example Kingma & Welling (2013); Rezende et al. (2014)), the variational distribution $q_\phi(z|y)$ is only being used to tighten the resulting bound and is not a component of the generative model, although it can be used to infer structure in the latent space.

As an example, the above provides the following upper bound on the reverse KL divergence[2]

$$\mathrm{KL}(p_\theta(y)||p(y)) \leq \int p_\theta(y|z)p(z)\left[\log p_\theta(y|z)p(z) - \log q_\phi(z|y)p(y)\right]dydz \qquad (11)$$

The upper bound (11) can now be estimated through sampling and minimized with respect to $\theta$ and $\phi$ by using the reparameterization trick (Kingma & Welling (2013)) and taking gradients. This results in a tractable procedure to minimize $\mathrm{KL}(p_\theta(y)||p(y))$[3]. Note that this will require the estimation of the gradient $\nabla_\theta \int p_\theta(y|z)p(z) \log p(y)dxdz$, for which we propose an efficient (approximately unbiased) gradient estimator in the supplementary materials section B.

For $f$-divergences other than the reverse KL, since the joint $f$-divergence $\mathrm{D}_f(p(y,z)||p_\theta(y,z))$ is expressed as an expectation over $p_\theta(y|z)p(z)$, we can use a similar procedure to optimize the upper bound (10). Namely we can generate $(y,z)$ samples from these distributions and then estimate the bound and take gradients.

## 4 EXPERIMENTS

In the following experiments our interest is to demonstrate the applicability of the $f$-divergence upper bound. The main focus is on training with the reverse KL divergence since this provides a natural "opposite" to training with the forward KL divergence . Throughout, the data is continuous and we use a Gaussian noise process with width $\sigma$ for $p(y|x)$. We take $p(z)$ to be a standard zero mean unit covariance Gaussian (thus with no trainable parameters). Similar to standard VAE training, we use deep networks to parameterize the Gaussian model $p_\theta(y|z) = \mathcal{N}(y|\mu_\theta(z), \sigma^2)$ and Gaussian variational distribution $q_\phi(z|y) = \mathcal{N}(z|\mu_\phi(y), \Sigma_\phi(y))$ for diagonal $\Sigma_\phi(y)$. Experimentally, we found that running several optimizer steps on $\phi$ whilst keeping $\theta$ fixed is also useful to ensure that the bound is tight when adjusting $\theta$. We therefore use this strategy throughout training. The result that

---

[2]As for the general $f$-divergence, we write the bound for continuous $x$ but there is a natural analogue for discrete $x$. One simply replaces integration with summation.

[3]Note that the optimization of the reverse KL in the standard $x$ space is intractable since we cannot obtain an unbiased estimate for the entropy of $p_\theta(x)$. Whilst a biased estimate could be attempted by sampling, this would require a nested sampling strategy, making this computationally unfeasible.

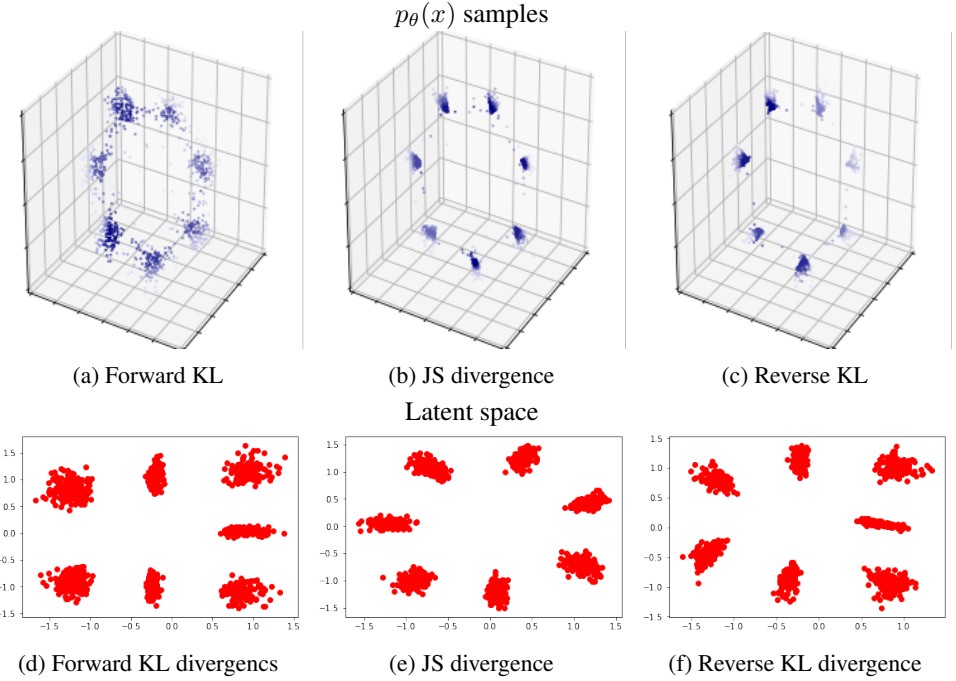

Figure 3: Toy problem. In (a), (b) and (c) we plot the samples of the model $p_\theta(x)$ trained by three different $f$-divergences. In (d), (e) and (f) we plot the latent $z$ by sampling from the trained $q_\phi(z|x)$ for each datapoint $x$. Note that this is after inverting the noise process, to recover the model on the $x$ space. See also section C

optimizing the auxiliary bound with respect to only $q_\phi(z|y)$ tightens the bound (towards the marginal divergence) is shown in the supplementary materials section A.

### 4.1 TOY PROBLEM : FOWARD KL, REVERSE KL AND JS TRAINING

The toy dataset, as described by Roth et al. (2017), is a mixture of seven two-dimensional Gaussians arranged in a circle and embedded in three dimensional space, see figures: figure 3, figure 6. We use 5 hidden layers of 400 units and relu activation function for the mean and variance parameterization in $q_\phi(z|y)$ and mean parameterization in $p_\theta(y|z)$ with a two dimensional latent space $z \in \mathbb{R}^2$.

We use the KL (moment matching) objective $KL(p(y)||p_\theta(y))$, reverse KL (mode seeking) objective $KL(p_\theta(y)||p(y))$ objective, and JS divergence (balance between KL and reverse KL) objective $JS(p_\theta(y)||p(y))$ to train the model. We minimize the corresponding auxiliary $f$-divergence bounds by gradient descent using the Adam optimizer (Kingma & Ba, 2014) with learning rate $10^{-3}$.

To evaluate the bound in each iteration we use a minibatch of size 100 to calculate $p^{(B)}(y)$. For each minibatch we draw 100 samples from $p(z)$ and subsequently draw 10 samples from $p_\theta(y|z)$ for each drawn $z$ to generate $(y, z)$ samples. To facilitate training, we anneal the width, $\sigma$, of the spread divergence throughout the optimization process. This enables the model to feel the presence of other distant modes (high mass regions of $p(y)$), allowing the method to overcome any poor initialization of $(\theta, \phi)$. In this experiment, $\sigma$ is annealed from $1.0$ to $0.1$ using the formula $\sigma = 1.0 * (0.1^{\frac{current\ steps}{total\ steps}})$.

We can see in figure 3 that the model trained by JS and reverse KL divergence converge to cover each mode in the true generating distribution, and exhibits good separation of the seven modes in the latent space. Even though the reverse KL tends to collapse a model to a single mode, provided the model $p_\theta(y|z)$ is sufficiently powerful, it can correctly capture all the 7 modes.

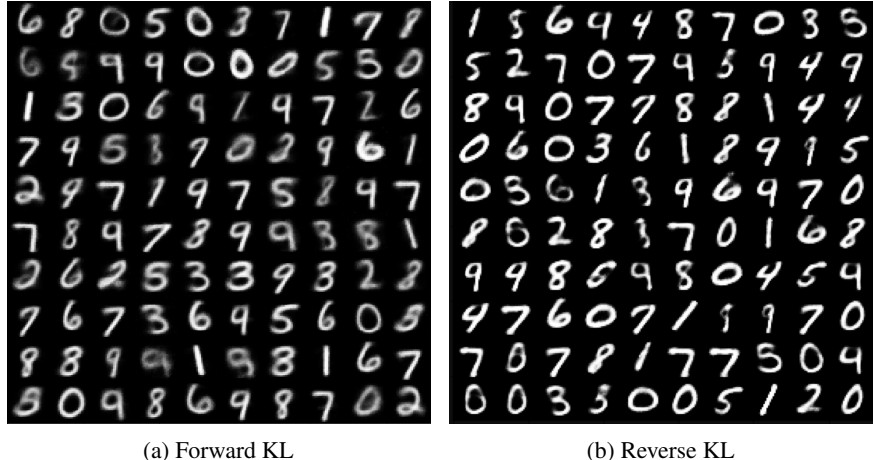

(a) Forward KL                                    (b) Reverse KL

Figure 4: MNIST experiment. (a) Samples from the models trained by forward KL (b) Samples from the models trained by reverse KL.

## 4.2 MNIST : FORWARD AND REVERSE KL TRAINING

To model the standard MNIST handwritten character dataset (LeCun & Cortes (2010)), we parametrize the mean of $p_\theta(y|z)$ by a neural network and the variance of the spread divergence is fixed to 0.5 for both RKL training and forward KL training. For $q_\phi(z|y)$ we use a Gaussian with mean and isotropic covariance parameterized by a neural network. Both networks contain 5 layers, each layer with 400 units and leaky-relu as activation function. The latent $z$ has dimension 20. We train both forward KL and reverse KL divergence using the $f$-divergence upper bound. For the reverse KL experiment, we first initialize the model by training using the forward KL objective for 10 epochs to prevent getting a sub-optimal solution (e.g. mode collapsing). After that, we train the model using the reverse KL objective for additional 10 epochs. We interleave each $\theta$ update (learning rate $10^{-4}$ with SGD) with 20 $\phi$ updates (learning rate $10^{-3}$ with Adam). We use the Monte Carlo estimation of the gradient from $\log p(y)$ (discussed in section B).

We also train the model by forward KL for 20 epochs to make a comparison. All the networks are trained using the Adam optimizer with learning rate $10^{-4}$. In figure 4 we show the samples from two models. As we can see, model trained by reverse KL generates sharper images than the model trained by forward KL.

## 4.3 CELEBA : FORWARD AND REVERSE KL TRAINING

We pre-process CelebA (Liu et al., 2015) images by first taking 140x140 center crops and then resizing to 64x64. Pixel values were then rescaled to lie in $[0, 1]$. The architectures of the convolutional encoder $q_\phi(z|y)$ and deconvolutional decoder $p_\theta(y|z)$ (with fixed variance 0.5) are given in the supplementary material, section D. The standard deviation of the spread divergence is 0.5. We train a VAE for 2 epochs as initialization and then train for one additional epoch for both pure forward KL and reverse KL. In order to ensure that the bound remained tight, we interleave each $\theta$ update (learning rate $10^{-5}$ with SGD) with 20 $\phi$ updates (learning rate $10^{-5}$ with Adam), training used in both cases.

In figure 5 we show samples from the trained models. As we can see, the impact of the reverse KL term in training is significant, resulting in less variability in pose, but sharper images. This is consistent with the "mode-seeking" behavior of the reverse KL objective.

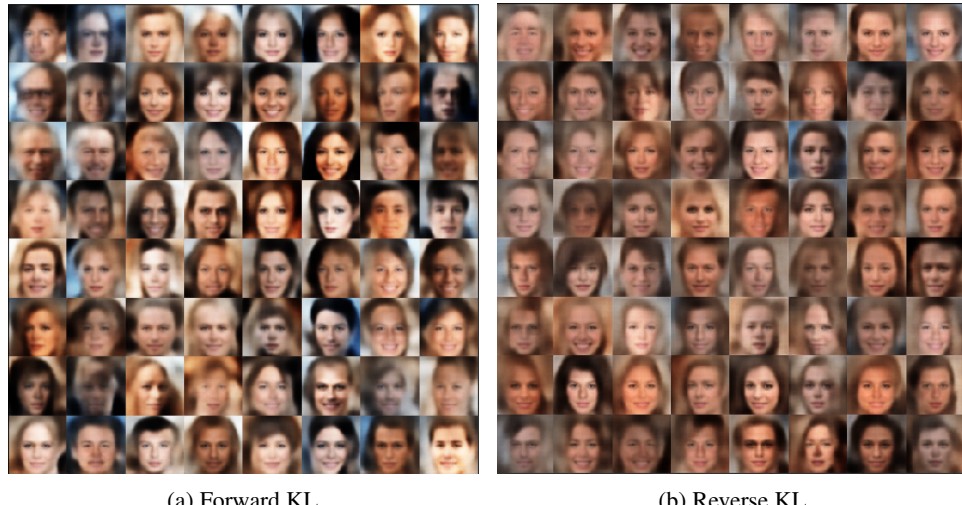

| (a) Forward KL | (b) Reverse KL |

Figure 5: CelebA experiment. Image samples from the trained models $p_\theta(x)$. After VAE initialization, we continued training for an additional epoch with (a) pure forward KL and (b) the reverse KL divergence.

## 4.4 $f$-GAN COMPARISON

As discussed in section 2.3, a different approach to minimizing the $f$-divergence is used in Nowozin et al. (2016), utilizing a variational *lower bound* to the $f$-divergence:

$$D_f(p(x)||p_\theta(x)) \geq \sup_{T\in\mathcal{T}} \left( \mathbb{E}_{x\sim p(x)}[T(x)] - \mathbb{E}_{x\sim p_\theta(x)}[f^*(T(x))] \right) \tag{12}$$

Here $f^*$ is the Fenchel conjugate and $\mathcal{T}$ is any class of functions that respects the domain of $f^*$. After parameterizing $T = g_f(V_\phi)$ (where $g_f : \mathbb{R} \to dom_{f^*}$ and $V_\phi$ is an unconstrained parametric function) and $p_\theta(x)$, the optimization scheme is then to alternately tighten (i.e. increase) the bound through changes to $\phi$ and then lower the bound through changes to $\theta$, see figure 2. This is of interest because the GAN objective (Goodfellow et al., 2014) can be seen as a specific instance of this scheme. We acknowledge that Nowozin et al. (2016) principally grounds GANs in a wider class of techniques, and is not necessarily intended as a scheme for minimizing an $f$-divergence. However, it is natural to ask whether our auxiliary upper bound or the Fenchel-conjugate lower bound give different results when used to minimize the $f$-divergence for a similar complexity of parameter space $(\theta, \phi)$.

To compare the two methods we fit a univariate Gaussian $p_\theta(x)$ to data generated from a mixture of two Gaussians through the minimization of various $f$-divergences. See the supplementary material for details. For the $f$-GAN lower bound we use a network with two hidden layers of size 64 for $V_\phi(x)$. For our upper bound we use a network with two hidden layers of size 50 to parameterize $q_\phi(z|x)$ and set $p_\theta(x, z)$ to be a bivariate Gaussian, so that it marginalizes to a univariate Gaussian as required. The upper and lower bound methods have a similar number of free parameters ($q_\phi$ has fewer hidden units but more outputs than $V_\phi$). The two methods result in broadly similar Gaussian fits, see table 2. In general, minimizing the upper bound results in a slightly superior fit compared to the $f$-GAN method (Nowozin et al., 2016) in terms of proximity to the true minimal $f$-divergence fit and proximity of the bound value to the true divergence. Additionally we find that minimizing our upper bound is computationally more stable than the optimization procedure required for $f$-GAN training (simultaneous tightening and lowering of the bound – see supplementary material E).

## 5 RELATED WORK

The *Auxiliary Variational Method* (Agakov & Barber, 2004) uses an auxiliary space to minimize the joint KL divergence in order to minimize the marginal KL divergence. We extend this method to the more general class of $f$-divergences.

|  | KL | rev-KL | J-S |
|---|---|---|---|
| $D_f(p(x)\|p_*(x))$ | 0.21 | 0.18 | 0.05 |
| $D_f(p(x)\|p_{LB}(x))$ | 0.32 | 0.25 | 0.23 |
| $D_f(p(x)\|p_{UB}(x))$ | **0.21** | **0.23** | **0.15** |
| $\mu^*$ | 1.70 | 1.85 | 1.76 |
| $\hat{\mu}_{LB}$ | 1.71 | 1.73 | 1.70 |
| $\hat{\mu}_{UB}$ | 1.71 | **1.76** | 1.70 |
| $\sigma^*$ | 0.62 | 0.57 | 0.60 |
| $\hat{\sigma}_{LB}$ | 0.46 | 0.45 | 0.24 |
| $\hat{\sigma}_{UB}$ | **0.62** | **0.65** | **0.33** |

Table 2: Learned Gaussian parameters to fit a mixture of two Gaussians using forward KL, reverse KL and Jensen-Shannon divergence. $p_*(x)$ is the optimal Gaussian fitted to minimize the exact divergence. $p_{UB}(x)$ is the optimal Gaussian fitted to minimize our auxiliary upper bound on the divergence. $p_{LB}(x)$ is the optimal Gaussian fitted to minimize the Fenchel-conjugate lower bound on the divergence.

*Variational Auto-Encoders* (Kingma & Welling, 2013) Our method is a way to train an identical class of generative and variational models, but with a class of different optimization objectives based on $f$-divergences. Since the VAE optimization scheme is a variational method of maximizing the likelihood, it is similar to our scheme with the choice of minimizing the forward KL divergence, which is also a variational form of maximum likelihood. Both methodologies use sampling to estimate a variational bound which can be differentiated through the use of the reparameterization trick.

In *Rényi divergence variational inference* (Li & Turner, 2016), a variational lower bound of log-likelihood is proposed based on the Rényi divergence. However, our joint upper bound is an estimator of $f$-divergence in marginal data space, it only relates to maximum likelihood learning when we use KL divergence.

In *Auxiliary Deep Generative Models* (Maaløe et al., 2016), a VAE is extended with an auxiliary space. This allows a richer variational distribution to be learned, with the correlation between latent variables being pushed to the auxiliary space to keep the calculation tractable. This, similarly to our method, utilizes the general auxiliary variational method Agakov & Barber (2004), but is focused on making VAEs more powerful rather than providing different optimization schemes.

In the $f$-GAN (Nowozin et al., 2016) methodology, an interesting connection is made between the GAN training objective and a lower bound on the $f$-divergence. The authors conclude that using different divergences leads to largely similar results, and that the divergence only has a large impact when the model is "under-powered". However, that conclusion is somewhat at odds with our own, in which we find that the (upper bound on) different divergences gives very different model fits. Indeed, others have reached a similar conclusion: the reverse KL divergence is optimized as a GAN objective in Sønderby et al. (2017), demonstrating that it is effective in the task of image super-resolution. A variety of different generator objectives for GANs are used in Poole et al. (2016), with some divergence objectives exhibiting the "mode-seeking" behavior we have observed.

In Mohamed & Lakshminarayanan (2016), the authors demonstrate an alternative approach to train $f$-divergence $D_f(p(x)\|q(x)) = \int q(x)f\left(\frac{p(x)}{q(x)}\right) dx$ by directly estimating the density ratio $\frac{p(x)}{q(x)}$. This method makes a connection to GANs: the discriminator is trained to approximate the ratio and the generator loss is designed based upon different choices of $f$-divergence (see the supplementary material section F for details). We thereby recognize there are three different tractable estimations of the $f$-divergence: 1. ratio estimation in the marginal space 2. Fenchel conjugate lower bound ($f$-GAN) and 3. the variational joint upper bound (introduced by our paper).

Ratio estimation by classification has also been extended to minimize the KL-divergence in the joint space (Huszár, 2017). Similarly, *Bi-directional GAN* (Donahue et al., 2016) and *Ali-GAN* (Dumoulin et al., 2016) augment the GAN generator with an additional inference network. Although these models focus on similar training objectives to our own, the purpose of using the joint space is different to that of our approach. Our method uses the joint distribution to create an upper bound in order to estimate the $f$-divergence in the marginal space; the latent representation is automatically achieved. In contrast, all three methods mentioned above expand the original space to a joint space just for learning the latent representation, the divergence is estimated by either ratio estimation or GAN approaches. Additionally, they only minimize the target divergence only at the limit of an optimal discriminator (or in the nonparametric limit, see Goodfellow et al. (2014) and Mescheder et al. (2017)), which may cause instability in the GAN training process (Arjovsky & Bottou, 2017).

## 6 CONCLUSION

We introduced an upper bound on $f$-divergences, based on an extension of the auxiliary variational method. The approach allows variational training of latent generative models in a much broader set of divergences than previously considered. We showed that the method requires only a modest change to the standard VAE training algorithm but can result in a qualitatively very different fitted model. For our low dimensional toy problems, both the forward KL and reverse KL can be effective in learning the model. However, for higher dimensional image generation, compared to standard forward KL training (VAE), training with the reverse KL tends to focus much more on ensuring that data is generated with high fidelity around a smaller number of modes. The central contribution of our work is to facilitate the application of more general $f$-divergences to training of probabilistic generative models with different divergences potentially giving rise to very different learned models.

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

SUPPLEMENTARY MATERIAL

## A  PROPERTIES OF THE AUXILIARY VARIATIONAL METHOD

Here we give a property of the auxiliary bound for $f$-divergences with differentiable $f$; this covers most $f$ of interest, and the argument extends to those $f$ which are piecewise differentiable. Then for the particular case of the reverse Kl divergence we give a simpler proof of this property as well as two additional properties (which do not hold for general $f$).

### A.1  $D_f(p(x)||q(x))$

For differentiable $f$ we claim that when we fully optimize the auxiliary $f$-divergence w.r.t $p(z|x)$, this is the same as minimizing the $f$-divergence in the $x$ space alone.

Let's first fix $q(x, z)$ and find the optimal $p(z|x)$ by taking the functional derivative of the auxiliary $f$-divergence

$$\frac{\delta}{\delta p(z|x)} D_f(P||Q) = \frac{\delta}{\delta p(z|x)} \int q(x', z') f\left(\frac{p(z'|x')p(x')}{q(x', z')}\right) dx'dz' \tag{13}$$

$$= q(x, z) f'\left(\frac{p(z|x)p(x)}{q(x, z)}\right) \frac{p(x)}{q(x, z)} \tag{14}$$

$$= p(x) f'\left(\frac{p(z|x)p(x)}{q(z|x)q(x)}\right) \tag{15}$$

At the minimum this will be equal to 0 (plus a constant Lagrange multiplier that comes from the constraint that $p(z|x)$ is normalized). Since $f'$ is not constant (if it is then the $f$-divergence is a constant), this then implies that the argument of $f'$ must be constant in $z$. This implies that optimally $p(z|x) = q(z|x)$. Plugging this back into the $f$-divergence, it reduces to simply $D_f(p(x)||q(x))$

Hence, we have shown

$$\min_{p(z|x)} D_f(p(x, z)||q(x, z)) = D_f(p(x)||q(x)) \tag{16}$$

Since the assumption is that $D_f(p(x)||q(x))$ is not computationally tractable, this means that, in practice, we need to use a suboptimal $p(z|x)$, restricting $p(z|x)$ to a family $p_\theta(z|x)$ such that the joint $f$-divergence is computationally tractable.

### A.2  RELATION TO $KL(q(x)||p(x))$

For the particular case of the reverse KL divergence we also provide this more straightforward proof.

Again, the claim is that when we fully optimize the auxiliary KL divergence w.r.t $p(z|x)$, this is the same as minimizing the KL in the $x$ space alone.

Let's first fix $q(x, z)$ and find the optimal $p(z|x)$. The divergence is

$$KL(q(x, z)||p(x, z)) = -\int q(z|x)q(x) \log p(z|x) dxdz + const.$$

$$= \int q(x) KL(q(z|x)||p(z|x)) \, dx + const. \tag{17}$$

Since we are taking a positive combination of KL divergences, this means that, optimally, $p(z|x) = q(z|x)$. Plugging this back into the KL divergence, the KL reduces to simply

$$KL(q(x)||p(x)) \tag{18}$$

Hence, we have shown

$$\min_{p(z|x)} KL(q(x, z)||p(z|x)p(x)) = KL(q(x)||p(x)) \tag{19}$$

### A.3   INDEPENDENCE $p(z|x) = p(z)$

Also for the particular case of the reverse KL divergence we can derive a result from the assumption that the auxiliary variables are independent of the observations and the prior $p(z|x) = p(z)$. We have

$$
\begin{aligned}
\mathrm{KL}(q(x,z)||p(x,z)) &= \int q(x,z)\log q(x,z)dxdz - \int q(x,z)\log[p(z)p(x)]dxdz \\
&= \mathrm{KL}(q(z)||p(z)) + \int q(z)\mathrm{KL}(q(x|z)||p(x))\,dz
\end{aligned}
\tag{20}
$$

Optimally, therefore, we set $p(z) = q(z)$, which gives the resulting expression

$$
\int_z q(z)\mathrm{KL}(q(x|z)||p(x))
\tag{21}
$$

Since we are still free to set $q(z)$, we should optimally set $q(z)$ to place all its mass on the $z$ that minimizes

$$
\mathrm{KL}(q(x|z)||p(x))
\tag{22}
$$

In other words, the assumption of independence $p(z|x) = p(z)$ implies that method is no better than computing each $\mathrm{KL}(q(x|z)||p(x))$ and then choosing the single best model $q(x|z)$.

### A.4   FACTORIZING $q(x,z) = q(x)q(z)$

Again for the reverse KL divergence, under the independence assumption $q(x,z) = q(x)q(z)$, it is straightforward to show that

$$
\mathrm{KL}(q(x,z)||p(x,z)) = \mathrm{KL}(q(x)||p(x))
\tag{23}
$$

In the case that $q(x)$ for example is a simple Gaussian distribution, this means that the independence assumption does not help enrich the complexity of the approximating distribution.

### A.5   RELATION TO THE ELBO

The reverse KL divergence in joint space: $\mathrm{KL}(q(x,z)||p(x,z))$ is equivalent to using the ELBO to lower bound $\log p(x)$ in $\mathrm{KL}(q(x)||p(x))$:

$$
\begin{aligned}
\mathrm{KL}(q(x)||p(x)) &= \int q(x)(\log q(x) - \log p(x))dx \\
&\leq \int q(x)\left(\log q(x) - \underbrace{\int q(z|x)(\log p(x,z) - \log q(z|x))\,dz}_{\mathrm{ELBO}}\right)dx \\
&= \int q(x)q(z|x)(\log q(x)q(z|x) - \log p(x,z))dxdz \\
&= \mathrm{KL}(q(x,z)||p(x,z))
\end{aligned}
\tag{24}
$$

## B   APPROXIMATING GRADIENTS OF $\log p(y)$

For the reverse KL upper bound, we need to take calculate $\nabla_\theta \int p_\theta(y|z)p(z)\log p(y)dxdz$, where $p(y) = N^{-1}\sum_m p(y|x^{(n)})$, i.e. a sum of delta functions (the data distribution) corrupted with a noise process as described in section 3.1.

Clearly, summing over all points in the dataset to calculate $p(y)$ is computationally burdensome. Naively using a minibatch to estimate $p(y)$ inside the log results in a biased estimator, $\nabla_x \log\left(M^{-1}\sum_n p(y|x^{(m)})\right)$, which we have found to be detrimental to the optimization procedure in our experiments.

To proceed, let us assume that $p_\theta(y|z)$ and $p(y|x^{(n)})$ are spherical Gaussians with the same variance $\sigma^2$ (which is the case in our experiments)

$$\nabla_\theta \int p_\theta(y|z)p(z)\log p(y)dydz \tag{25}$$

$$= \nabla_\theta \int \frac{1}{(2\pi\sigma^2)^{D/2}} \exp\left(-\frac{1}{2\sigma^2}(y-\mu_\theta(z))^2\right) p(z)\log\left(N^{-1}\sum_n p(y|x^{(n)})\right)dydz \tag{26}$$

$$= \int p(\epsilon)p(z)\nabla_\theta \log\left(\frac{1}{N(2\pi\sigma^2)^{D/2}}\sum_n \exp\left(-\frac{1}{2\sigma^2}\left(\mu_\theta(z)+\sigma\epsilon-x^{(n)}\right)^2\right)\right)d\epsilon dz \tag{27}$$

$$= -\frac{1}{\sigma^2}\int p(\epsilon)p(z)\nabla_\theta\mu_\theta(z)\frac{\sum_n\left(\mu_\theta(z)+\sigma\epsilon-x^{(n)}\right)\exp\left(-\frac{1}{2\sigma^2}\left(\mu_\theta(z)+\sigma\epsilon-x^{(n)}\right)^2\right)}{\sum_n \exp\left(-\frac{1}{2\sigma^2}\left(\mu_\theta(z)+\sigma\epsilon-x^{(n)}\right)^2\right)}d\epsilon dz \tag{28}$$

$$= -\frac{1}{\sigma^2}\int p(\epsilon)p(z)\nabla_\theta\mu_\theta(z)\sum_n\left(\mu_\theta(z)+\sigma\epsilon-x^{(n)}\right)p(n|\epsilon,z)d\epsilon dz \tag{29}$$

Where we have used the reparametrization $y = \mu_\theta(z) + \sigma\epsilon$ and $p(\epsilon) = N(0,I)$, and then noticed that we can define

$$p(n|\epsilon,z) := \frac{\exp\left(-\frac{1}{2\sigma^2}\left(\mu_\theta(z)+\sigma\epsilon-x^{(n)}\right)^2\right)}{\sum_n \exp\left(-\frac{1}{2\sigma^2}\left(\mu_\theta(z)+\sigma\epsilon-x^{(n)}\right)^2\right)} \tag{30}$$

which is a softmax over the square distance (with a scaling) between $y$ and $x^{(n)}$.

We can now get an unbiased estimator for this gradient (29) if we can generate samples from $p(n|\epsilon,z)$.

The computationally expensive part of calculating $p(n|\epsilon,z)$ is the normalizer, which requires summing over all data points. Given that typically the $x$-space will be high dimensional in practice we consider a dimensionality reduction technique to speed up computing the square distance between $y$ and $x^{(n)}$.

We use Principal Components Analysis (PCA) to project the $x$-space to a much lower dimensional space. PCA is an appropriate choice as it maximizes the variance preserved by the lower dimensional projections, whilst minimizing the square distance between the reconstructions and the original data. Note that the PCA projection matrix, $U$, is learned once on the input data $\{x^{(n)}\}$.

So we approximate

$$p(n|\epsilon,z) \approx q(n|\epsilon,z) := \frac{\exp\left(-\frac{1}{2\sigma^2}\left(U^T(\mu_\theta(z)+\sigma\epsilon)-U^T(x^{(n)})\right)^2\right)}{\sum_n \exp\left(-\frac{1}{2\sigma^2}\left(U^T(\mu_\theta(z)+\sigma\epsilon)-U^T(x^{(n)})\right)^2\right)} \tag{31}$$

We can now get an (approximate) unbiased estimator for (29) by sampling $\epsilon,z$ and then $n \sim q(n|\epsilon,z)$.

$$\nabla_\theta \int p_\theta(y|z)p(z)\log p(y)dydz \approx -\frac{1}{S\sigma^2}\sum_{s=1}^S\left[\nabla_\theta\mu_\theta(z^{(s)})\frac{1}{T}\sum_{t=1}^T\left(\mu_\theta(z^{(s)})+\sigma\epsilon^{(s)}-x^{(n_s^{(t)})}\right)\right] \tag{32}$$

Where $z^{(s)} \sim p(z)$, $\epsilon^{(s)} \sim p(\epsilon)$, and for each s we sample $n_s^{(t)} \sim q(n|\epsilon^{(s)},z^{(s)})$. In the experiments on MNIST/CelebA, we sample $T = 10$ to form the approximation, and the PCA dimension is 50. We also find that scaling the contribution from this term to the gradient can sometimes improve optimization (we use 0.2 as the scaling factor in the image generation experiments).

Using this approximation, the computation cost of the normalizer in (32) scales with the number of the data points, but we have found this not to be an issue in practice when using the PCA projection.

For a very large dataset this could be problematic though. In this case other minibatch methods could be used to approximate this normalizer, such as Ruiz et al. (2018) and Botev et al. (2017), which we leave to further work.

## C  TARGET DISTRIBUTION OF THE TOY PROBLEM

We train on the toy dataset described by Roth et al. (2017), which is a mixture of seven two-dimensional Gaussians arranged in a circle and embedded in three dimensional space, see figure 6. The standard deviation of the each Gaussian is 0.05.

## D  NETWORK ARCHITECTURE

Both encoder and decoder used fully convolutional architectures with 5x5 convolutional filters and used vertical and horizontal strides 2 except the last deconvolution layer we used stride 1. Here $\text{Conv}_k$ stands for a convolution with $k$ filters, $\text{DeConv}_k$ for a deconvolution with k filters, BN for the batch normalization Ioffe & Szegedy (2015), ReLU for the rectified linear units, and $\text{FC}_k$ for the fully connected layer mapping to $R^k$.

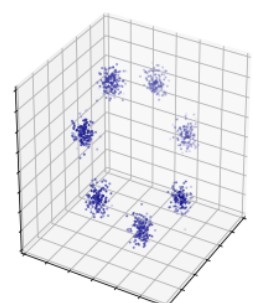

Figure 6: Target distribution of the toy problem, from Roth et al. (2017)

$$x \in R^{64 \times 64 \times 3} \to \text{Conv}_{128} \to \text{BN} \to \text{Relu}$$
$$\to \text{Conv}_{256} \to \text{BN} \to \text{Relu}$$
$$\to \text{Conv}_{512} \to \text{BN} \to \text{Relu}$$
$$\to \text{Conv}_{1024} \to \text{BN} \to \text{Relu} \to \text{FC}_{64}$$

$$z \in R^{64} \to \text{FC}_{8 \times 8 \times 1024}$$
$$\to \text{DeConv}_{512} \to \text{BN} \to \text{Relu}$$
$$\to \text{DeConv}_{256} \to \text{BN} \to \text{Relu}$$
$$\to \text{DeConv}_{128} \to \text{BN} \to \text{Relu}$$
$$\to \text{DeConv}_{64} \to \text{BN} \to \text{Relu} \to \text{DeConv}_3$$

## E  $f$-GAN COMPARISON

The mixture of Gaussians we attempt to fit a univariate Gaussian to is plotted in Figure 7.

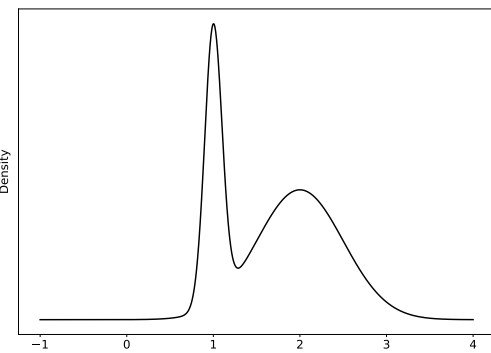

Figure 7: Mixture of two Gaussians, $0.3\mathcal{N}_1 + 0.7\mathcal{N}_2$ where $\mathcal{N}_1 = \mathcal{N}(\mu_1 = 1, \sigma_1 = 0.1)$ and $\mathcal{N}_2 = \mathcal{N}(\mu_2 = 2, \sigma_2 = 0.5)$

We plot the lower and upper bounds during training in Figure 8. We can see the upper bound is generally faster to converge and less noisy. It also a consistently decreasing objective, whereas the variational lower bound fluctuates higher and lower in value throughout the training process.

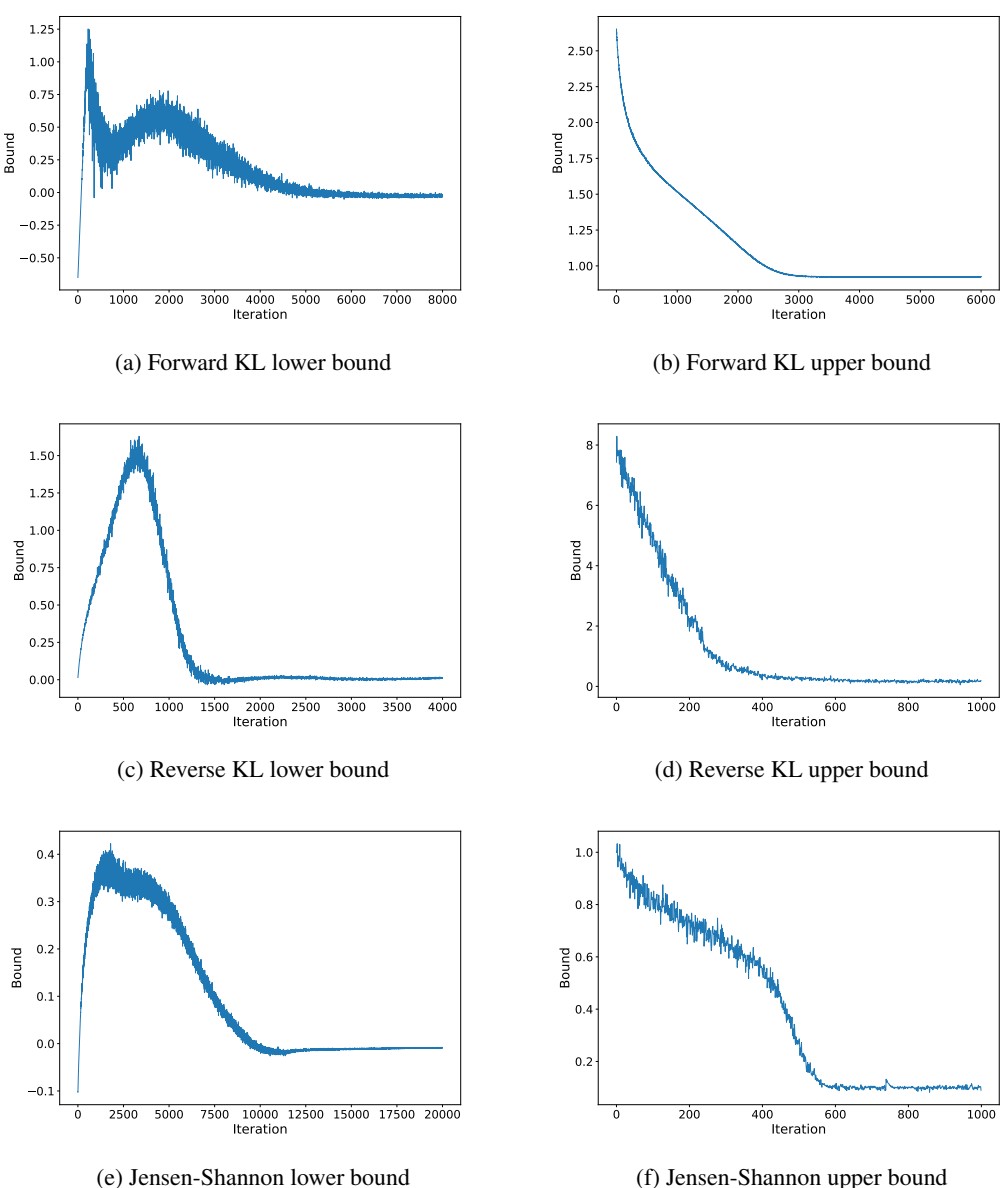

(a) Forward KL lower bound

(b) Forward KL upper bound

(c) Reverse KL lower bound

(d) Reverse KL upper bound

(e) Jensen-Shannon lower bound

(f) Jensen-Shannon upper bound

Figure 8: Training runs for fitting a univariate Gaussian to a mixture of two Gaussians by minimizing a variety of $f$-divergences. On the left we train using the lower bound, on the right with our upper bound.

# F    CLASS PROBABILITY ESTIMATION

In Mohamed & Lakshminarayanan (2016), two ratio estimation techniques, class probability estimation and ratio matching, are discussed. We briefly show how to use the class probability estimation technique to estimate $f$-divergence, and refer readers to the original paper Mohamed & Lakshminarayanan (2016) for the ratio matching technique.

The density ratio can be computed by building a classifier to distinguish between training data and the data generated by the model. This ratio is $\frac{p(x)}{q_\theta(x)} = \frac{p(x|y=1)}{p(x|y=0)}$, where label $y = 1$ represents samples from $p$ and $y = 0$ represents samples from $q$. By using Bayes rule and assuming that we have the same number of samples from both $p$ and $q$, we have $\frac{p(x)}{q_\theta(x)} = \frac{p(x|y=1)}{p(x|y=0)} = \frac{p(y=1|x)p(x)}{p(y=1)} / \frac{p(y=0|x)p(x)}{p(y=0)} = \frac{p(y=1|x)}{p(y=0|x)}$. We can then set the discriminator output to be $\mathcal{D}_\phi(x) = p(y = 1|x)$, so the ratio can be written as $\frac{p(x)}{q_\theta(x)} = \frac{p(y=1|x)}{1-p(y=1|x)} = \frac{\mathcal{D}_\phi(x)}{1-\mathcal{D}_\phi(x)}$. The generator loss corresponding to an $f$-divergence can then be designed as $D_f(p(x)||q_\theta(x)) = \int q_\theta(x)f\left(\frac{p(x)}{q_\theta(x)}\right) dx = \int q_\theta(x)f(\frac{\mathcal{D}_\phi(x)}{1-\mathcal{D}_\phi(x)}) = \mathbb{E}_{q(z)}[f(\frac{\mathcal{D}(\mathcal{G}_\theta(z))}{1-\mathcal{D}(\mathcal{G}_\theta(z))})]$.

