# OpenReview forum: "Training generative latent models  by variational f-divergence minimization"
_ICLR.cc/2019/Conference_

### Official Review · AnonReviewer1 · 2018-10-28
**The research direction itself is interesting, but further experimental validation is needed**

**Rating:** 5
**Confidence:** 3

**Review:**

\clarity & quality
The paper is easy to follow and self-contained.
However, the motivation for minimizing the upper bound is not so clear for me.
As far as I understood from the paper, changing the objective function to the upper bound of f-divergence have two merits compared to the existing methods. One is that by using the reverse KL, we can obtain sharper outputs, and the second one is that the optimization process will be stable compared to that of the lower bound.
In the introduction, the author just mentioned that "the f-divergence is generally computationally intractable for such complex models. The main contribution of our paper is the introduction of an upper bound on the f-divergence."
For me, this seems that the author just introduced the new fancy objective. I think the motivation to introduce the new objective function should be stated clearly.

\originality & significance
Although the upper bound of the f-divergence is the trivial extension, the idea to optimize the upper bound for the latent model seems new and interesting.

However, it is hard to evaluate the usefulness of the proposed method from the current experiments.
It seems that there are two merits about the proposed method as above.
The only evidence that the learning tends to be stable is the Fig.8 in the appendix, but this is just the fitting of univariate Gaussian to a mixture of Gaussians, thus it is too weak as the evidence.
About the sharp output, there are already many methods to overcome the blurred output of the usual VAE. No comparison is done in the paper.
So I cannot tell whether the proposed objective is really useful to learn the deep generative models.
I think further experimental results are needed to validate the proposed method.

\Question
In page 4,  the variance of the p(y|x) and p_\theta(y|z) are set to be the same. What is the intuition behind this trick?
Since this p(y|x) is used as the estimator for the log p(y) as the smoothed delta function whose Gaussian window width (the variance), and the Gaussian window width is crucial to this kind of estimator, I know why the author used this trick.

---

> ### Author Response · Authors · 2018-11-26
> **Response to reviewer 3**
>
> Thank you for the constructive feedback on our paper. We respond below to the points raised.
>
> > "For me, this seems that the author just introduced the new fancy objective. I think the motivation to introduce the new objective function should be stated clearly."
>
> We would argue that we haven't really introduced a new objective at all, but simply provided a tool to expand the scope of existing objectives to a wider family of models. Using probabilistic divergences as training objectives is a standard tool in generative modelling. Indeed the VAE is simply a special case of a general f-divergence objective.
>
> We do provide motivation for studying different training objectives in section 2. Simply put - different objectives result in different learned models, and so it can be of use to use a different objective depending on the desired use case for the model. For example if sharp samples are required then the reverse KL could be a suitable objective, whereas if good density estimation is required then the forward KL (maximum likelihood) may be a better objective.
>
> However, it is usually impossible to train a latent generative model that uses a divergence other than forward KL as the objective unless one uses a GAN style minimax optimization. The upper bound we propose in our paper both permits using any f-divergence as the objective, and is optimized via the minimization of an upper bound, which is more natural than the alternate tightening and minimizing of the Fenchel conjugate lower bound seen in the f-GAN optimization.
>
> > "The only evidence that the learning tends to be stable is the Fig.8 in the appendix, but this is just the fitting of univariate Gaussian to a mixture of Gaussians, thus it is too weak as the evidence."
>
> The comparison done to f-GAN training illustrates the advantage of minimizing an upper bound on the objective rather than alternately tightening and then minimizing a lower bound. This is only a toy comparison, but demonstrates that training is more straightforward in this case. For more complex examples, since we are minimizing an upper bound we have no reason to expect the training to be unstable.
>
> > "About the sharp output, there are already many methods to overcome the blurred output of the usual VAE. No comparison is done in the paper."
>
> Although there are ways to sharpen the output of VAEs , we are not aware of methods which are based on using a different divergence as the objective - which is a simple and principled way to affect the learned model.
>
> > "In page 4,  the variance of the p(y|x) and ptheta(y|z) are set to be the same. What is the intuition behind this trick? "
>
> Using the same variance for both distributions is as per the spread divergence methodology, as it allows us to invert the noise process easily. We have added to section 3.2 to make this clearer. For a deeper discussion see the spread divergence paper https://arxiv.org/abs/1811.08968.

---

### Official Review · AnonReviewer3 · 2018-11-01
**An interesting paper with weak experiments**

**Rating:** 5
**Confidence:** 4

**Review:**

This paper proposed a novel variational upper bound for f-divergence, one example of which is the famous evidence lower bound for max-loglikelihood learning. The second contribution might be the spread f-divergence for distributions having different supports. Even though theoretically sound, I believe that the presented experimental results are not strong enough to support the effectiveness of the proposed techniques. Detailed comments are listed below.

1) Notations are confusing, especially in Section 3 when introducing the SPREAD f -DIVERGENCE.
2) I cannot find on arXiv the reference “D. Barber, M. Zhang, R. Habib, and T. Bird. Spread divergences. arXiv preprint, 2018.” So I am not sure whether you can take credit from the “spread f-divergence” or not.
3) Important analysis/experiments on several key points are missing, for example, (i) how to specify the variance of the spread divergence in practice? (ii) how to estimate log p(y)? What is the influence?
4) In the paragraph before Sec 4.2, how the sigma of the spread divergence is annealed?
5) Despite the toy experiment in Sec 4.4, what are the advantages of the proposed f-divergence upper bound over the Fenchel-conjugate f-divergence lower bound? The current experimental results barely show any advantage.

Minors:
1) Page 6, under Figure 3. The statements of “KL (moment matching)” and “reverse KL (mode seeking)” are not consistent with what’s stated in Sec 2.2 (the paragraph under Eq (3)).
2) “RKL” and “JS” are not defined. Forward KL and standard KL are both used in the paper.

---

> ### Author Response · Authors · 2018-11-26
> **Response to reviewer 2**
>
> Thank you for the constructive feedback on the paper. We respond to your points below.
>
> > "2) I cannot find on arXiv the reference “D. Barber, M. Zhang, R. Habib, and T. Bird. Spread divergences. arXiv preprint, 2018.” So I am not sure whether you can take credit from the “spread f-divergence” or not."
>
> The paper has recently been uploaded to Arxiv. Here is the link: http://arxiv.org/abs/1811.08968
>
> > "3) Important analysis/experiments on several key points are missing, for example, (i) how to specify the variance of the spread divergence in practice? (ii) how to estimate log p(y)? What is the influence?"
>
> The width of the noise-corruption process, p(y|x), in the spread divergence is a hyper-parameter to be set and tuned like any other in the training process. We have given in all our experiments the values we used. Conceptually, the larger the width of the spread divergence then the more stable the training, as there will be fewer regions of low probability mass that can destabilize training. However, if the noise is too large it could "drown out" the signal from the training data (to illustrate, consider the extreme case of infinite noise).
>
> > "4) In the paragraph before Sec 4.2, how the sigma of the spread divergence is annealed?"
>
> We have added details in the third paragraph of Section 4.1 to clarify this point.
>
> > "5) Despite the toy experiment in Sec 4.4, what are the advantages of the proposed f-divergence upper bound over the Fenchel-conjugate f-divergence lower bound? The current experimental results barely show any advantage."
>
> This is an excellent question and at the core of our contribution. You are right that both methods aim to minimise an f-divergence but the Fenchel-conjugate achieves this by minimising a *lower bound*.
>
> Using a lower bound for minimization is a very unnatural thing to do as minimising the bound may simply make it looser. This leads to the need for an unstable min-max training procedure where one first tries to make the bound tight and then minimises. Perhaps the best justification for this unnatural procedure is that it works.
>
> What we hope to have shown is that one need not take this approach. Its perfectly possible to get good results by minimizing a more natural upper-bound on the f-divergence and this leads to more stable training.
>
> The toy comparison illustrates that our method slightly outperforms the lower bound optimization, though they are similar. The more important conclusion from our comparison is that, to achieve similar or even better results, one does not have to adopt the unstable minimax training procedure.

---

### Official Review · AnonReviewer2 · 2018-11-02
**Nice trick to utilize an arbitrary f-divergence as the objective for training generative models.**

**Rating:** 6
**Confidence:** 3

**Review:**

The paper proposes a method for training generative models with general f-divergences between the model and empirical distribution (the VAE and GAN objectives are captured as specific instantiations). The trick that leads to computational tractability involves utilizing a latent variable and optimizing the f-divergence between joint distributions which is an upper bound to the (desired) f-divergence between the marginal distributions. Distribution support issues are handled by convolving the data space with a blurring function. Empirical results on image datasets demonstrate that the additional training flexibility results in qualitatively different learned models. Specifically, optimizing the reverse KL (and/or Jensen-Shannon divergence) leads to sharper images, perhaps at the loss of some variance in the data distribution.

I liked the simplicity of the core idea, and appreciated the exposition being generally easy to follow. The application of upper bounds to general f-divergences for training generative models is novel as far as I know. My two issues are with the practicality of the implementation and evaluation methodology, both potentially affecting the significance of the work. Regarding practicality, it appears the training details are specific to each experiment. This begs the question of how sensitive the results are to these settings. Regarding the methodology, it would have been nice to see the method of Nowozin et al. (2016) applied in all experiments since this is a direct competitor to the proposed method. Moreover, the subjective nature of the results in the real dataset experiments makes it difficult to judge what the increased flexibility in training really provides - although I do note that the authors make this same point in the paper. Finally, given that model training is the paper's focus, an explicit discussion of computational cost was missed.

Even with these issues, however, I believe the paper makes a contribution to the important problem of fitting expressive generative models. In my opinion, a more rigorous and thorough experimental exploration would increase the value, but the paper demonstrates that training with alternative f-divergences is feasible.

---

> ### Author Response · Authors · 2018-11-26
> **Response to reviewer 1**
>
> We thank the reviewer for the thoughtful points raised.  We respond to your points below.
>
> > "Regarding practicality, it appears the training details are specific to each experiment. This begs the question of how sensitive the results are to these settings."
>
> The training details for both MNIST and the CelebA experiments are the same, aside from architectural and hyperparameter differences (which are to be expected). We interleave the same number of \phi updates between each \theta update, initialize with a partially trained VAE and use the same gradient estimator detailed in the supplementary materials section B (which has been added since the submission, and resulted in improved performance). The training details for the toy problem are simpler than for MNIST/CelebA, not requiring the VAE initialization or \phi interleaving. But we would argue that this is not problematic, as the toy problem does not pose the kind of optimization challenges seen in the more complex datasets.
>
> > "Regarding the methodology, it would have been nice to see the method of Nowozin et al. (2016) applied in all experiments since this is a direct competitor to the proposed method."
>
> We have performed a relatively simple comparison with the f-GAN method of Nowozin et al. in section 4.4. Although it is a toy comparison, it does show some salient differences between our training methods, in particular the beneficial effect of having an upper bound versus a lower bound for minimizing an objective. We have not included the f-GAN results for the other experiments we perform because the performance of the f-GAN is fairly well understood by the machine learning community.
>
> > "Finally, given that model training is the paper's focus, an explicit discussion of computational cost was missed."
>
> The main computational difference to VAE optimization results from the requirement to calculate gradients of log p(y). We propose an approximate unbiased Monte Carlo estimator for this gradient in the supplementary materials section B. Compared to the VAE, we need to calculate an additional normalizer for each sample in equation 31, whose complexity is O(T*N*D). T is the number of Monte Carlo samples to estimate the gradient, D is the dimension of the data after doing the PCA projection. N is the size of the training dataset, which is a potential computational bottleneck, but we find it is not slow in practice. In section B, we also refer several methods which can do faster unbiased Monte Carlo estimation of the normalizer by using minibatch methods, which we leave to future work.

---

### Meta-Review · Area_Chair1 · 2018-12-14
**Rejection: interest idea but empirical results are too week**

**Confidence:** 4
**Recommendation:** Reject

**Metareview:**

The paper proposes a new method for training generative models by minimizing general f-divergences. The main technical idea is to optimize f-divergence between joint distributions which is rightly observed to be the upper bound of the f-divergence between the marginal distributions and address the disjoint support problem by convolving the data with a noise distribution.  The basic ideas in this work are not completely novel but are put together in a new way.

However, the key weakness of this work, as all the reviewer noticed, is that the empirical results are too week to support the usefulness of the proposed approach. The only quantitive results are in table 2, which is only a simple Gaussian example. It essential to have more substantial empirical results for supporting the new algorithm.